# RQ-RAG: Learning to Refine Queries for Retrieval Augmented Generation

**Chi-Min Chan**[1]**, Chunpu Xu**[2]**, Ruibin Yuan**[1]**, Hongyin Luo**[3]**, Wei Xue**[1]**,
Yike Guo**[1]*,**Jie Fu**[1]*

[1] Hong Kong University of Science and Technology
[2] Hong Kong Polytechnic University
[3] Massachusetts Institute of Technology
zorowin123@gmail.com

## Abstract

Large Language Models (LLMs) exhibit remarkable capabilities but are prone to generating inaccurate or hallucinatory responses. This limitation stems from their reliance on vast pretraining datasets, making them susceptible to errors in unseen scenarios. To tackle these challenges, Retrieval-Augmented Generation (RAG) addresses this by incorporating external, relevant documents into the response generation process, thus leveraging non-parametric knowledge alongside LLMs' in-context learning abilities. However, existing RAG implementations primarily focus on initial input for context retrieval, overlooking the nuances of ambiguous or complex queries that necessitate further clarification or decomposition for accurate responses. To this end, we propose learning to Refine Queries for Retrieval Augmented Generation (RQ-RAG) in this paper, endeavoring to enhance the model by equipping it with capabilities for explicit rewriting, decomposition, and disambiguation. Our experimental results indicate that our method, when applied to a 7B Llama2 model, surpasses the previous state-of-the-art (SOTA) by an average of 1.9% across three single-hop QA datasets, and when applied to a 8B Llama3 model, it also demonstrates enhanced performance in handling complex, multi-hop QA datasets.

## 1 Introduction

Recent advancements in Large Language Models (LLMs) (OpenAI, 2023; Ouyang et al., 2022; Touvron et al., 2023) have demonstrated significant capabilities in understanding various concepts and solving downstream tasks (Brown et al., 2020; Raffel et al., 2020). Despite the vast amounts of data leveraged during their initial training or subsequent fine-tuning phases, these models inherently remain static. Once built and updated to a specific point in time, their knowledge base ceases to evolve, precluding the incorporation of new, real-time information. This limitation confines them to rely solely on their pre-encoded parametric knowledge (Mallen et al., 2022) during inference. Lacking access to up-to-date information, LLMs are prone to generating hallucinations (Ji et al., 2023) and may struggle to provide accurate and timely responses to queries that demand the latest information (Vu et al., 2023).

To overcome these challenges, integrating retrieval functionalities into generative models presents a promising solution (Lewis et al., 2020; Luo et al., 2023). This approach enriches standard parametric language models with non-parametric retrieval elements capable of accessing relevant information from external databases. These databases range from comprehensive document repositories, such as Wikipedia, to continuously updated sources like internet search engines, including Bing[1], DuckDuckGo[2], among others.

---

*Corresponding author
[1] bing.com
[2] duckduckgo.com

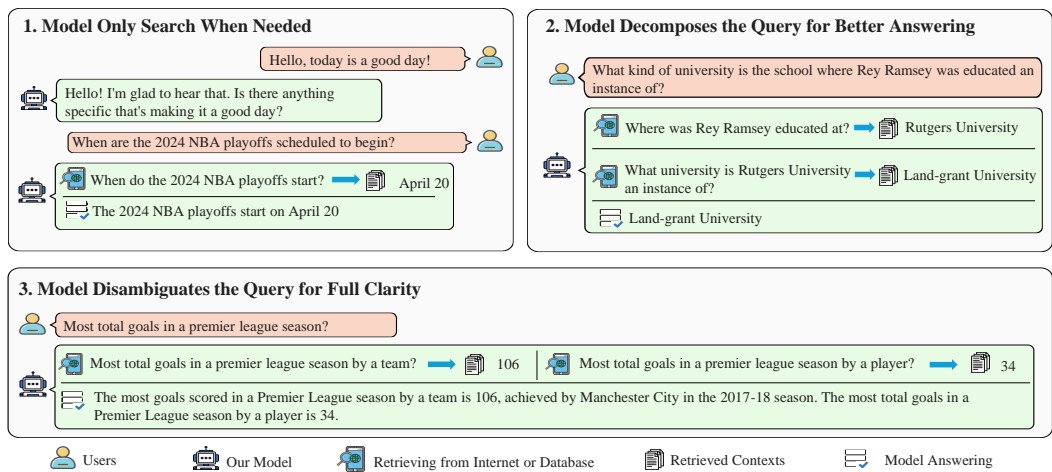

Figure 1: Our model are learned to *search on demand*, *rewrite*, *decompose* and *disambiguate* a query when needed.

Although such enhancements could mitigate the tendency towards inaccuracies and maintain the utility of LLMs in rapidly evolving domains, there still exist several issues in the previous framework. Firstly, the indiscriminate use of information retrieval systems to contextualize queries can be counterproductive. As demonstrated in previous research (Shi et al., 2023a), irrelevant context not only diminishes generation quality but may also obstruct LLMs' ability to answer queries they are otherwise capable of addressing. For straightforward queries like daily greetings, LLMs should respond directly rather than incorporating unnecessary context, thereby avoiding low-quality responses that could deter users. That is, the model should learn to search on demand, see Figure 1 (top-left). Secondly, for complex queries, simply searching with the original query often fails to retrieve adequate information. It's crucial for LLMs to first break down such queries into simpler, answerable sub-queries, and then search for information relevant to these sub-components. By integrating responses to these sub-queries, LLMs can construct a comprehensive answer to the original complex query. See Figure 1 (top-right). Lastly, for ambiguous queries with multiple possible answers, using the original query for information retrieval is insufficient. To provide complete and nuanced responses, LLMs must learn to clarify the query, ideally by identifying the user's intent, and then craft a more targeted search query. After collecting relevant information, LLMs can then deliver a detailed and comprehensive response, see Figure 1 (bottom).

Based on the above requirements, we propose **Learning to Refine Query for Retrieval Augmented Generation** (**RQ-RAG**) in this paper. We train a 7B Llama2 model in an end-to-end manner to enable it to dynamically refine search queries through rewriting, decomposing, and clarifying ambiguities. Our work draws inspiration from Self-RAG (Asai et al., 2024) and SAIL (Luo et al., 2023), which pioneer the augmentation of instructional tuning datasets with search results and teach models to filter out noise from search results for context-grounded response generation. Building on this foundation, we introduce innovative modifications to the process of crafting these datasets, enhancing the model's ability to produce more effective information retrievals. Specifically, we leverage ChatGPT[3] to craft tailored search queries across various scenarios (rewriting, decomposing, disambiguating) using distinct prompt templates, rather than relying on the original query. Furthermore, we observe instances where the dataset's initial output does not match the context returned by the information retrieval system. In these cases, we employ ChatGPT to generate new, contextually aligned answers, thereby enhancing the relevance and accuracy of the information retrieval process. Echoing the methodologies of prior research (Lu et al., 2022; Keskar et al., 2019; Asai et al., 2024), we employ control tokens—also known as special tokens—to

---

[3]By default, ChatGPT refers to **gpt-3.5-turbo-0125** and GPT-4 refers to **gpt-4-0125-preview** (OpenAI, 2023) in this paper.

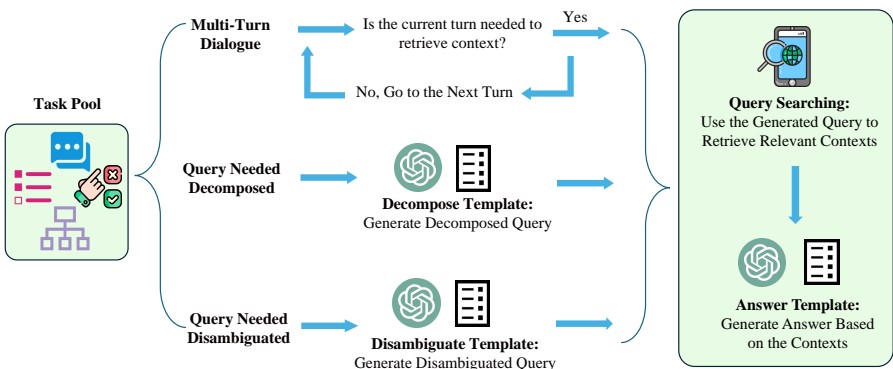

Figure 2: Dataset construction pipeline.

direct our generation process. With the application of multiple control tokens, our model can navigate through various trajectories in response to a user's query. At any given step, it has the flexibility to rewrite, decompose, disambiguate the query, or terminate the search process and proceed to generate responses.

To identify the optimal trajectory as our final answer, we meticulously developed three distinct selection methods that do not rely on external LLMs for trajectory evaluation (Yao et al., 2024). These methods include selection based on perplexity (PPL), confidence, and an ensemble approach, as elaborated in Section 2.3. Furthermore, we assess the upper bound of our method's performance by determining if any generated trajectory contains the correct answer. This analysis reveals a notably high potential for our system, underscoring its effectiveness if we can accurately select the correct trajectories.

To sum up, our paper makes several key contributions

- First, we show that 7B Llama2 model trained on our crafted dataset outperform previous state-of-the-art method (Asai et al., 2024) on three single hop QA tasks we evaluate, and also demonstrate superior performance on three multi-hop QA tasks, attributed to our query refinement approach.

- Second, we highlight the effectiveness of regenerating responses based on search results during data construction, moving beyond simply using the original dataset outputs. This method proves more effective than those employed in previous works (Luo et al., 2023; Asai et al., 2024), emphasizing the value of contextually grounded answer generation.

- Third, we showcase the great potential of our framework by illustrating its considerably high upper bound, as well as its resilience to different data sources compared to previous methods.

## 2 RQ-RAG: Learning to Refine Query for Retrieval Augmented Generation

In this section, we introduce RQ-RAG including the following part; Dataset Construction (Section 2.1), Generator Training (Section 2.2), and Sampling Strategies (Section 2.3).

### 2.1 Dataset Construction

To train the model to explicitly refine queries, the core of our pipeline involves collecting training data that mirrors the process at inference time. This includes generating refined queries for searches and crafting responses based on the information retrieved.

Given an input-output pair $(X_{\mathrm{origin}}, Y_{\mathrm{origin}})$ from the original dataset, our main objective is to construct a sequence of actions incorporating special tokens. These special tokens specify

the type of refinement $SPECIAL_{type}$ and is followed by the refined query, conditioned on the specific special token, which is represented as $Q_{i, type}$, where 'type' refers to the refinement action (either rewrite, decompose, or disambiguate) and 'i' refers to the turn of the iterations. Subsequently, we retrieve the top $k$ documents, denoted as $[D_{i1}, D_{i2}, \ldots, D_{ik}]$. At the final iteration step, we generate a new answer conditioned on the above contexts denoted as $Y_{new}$.

In short, our dataset construction can be seen as the transformation denoted as Equation 1.

$$(X_{origin}, Y_{origin}) \rightarrow (X_{origin}, \underbrace{SPECIAL_{type}, Q_{i, type}, [D_{i1}, \ldots, D_{ik}], \ldots}_{\text{repeat for i times}}, Y_{new}) \tag{1}$$

Furthermore, gathering a corpus that encompasses a wide range of scenarios, as outlined in Section 1, is crucial. Accordingly, our data collection focuses on scenarios that include multi-turn dialogue, queries requiring decomposition, and queries needing disambiguation. For a comprehensive breakdown of the data statistics, refer to Appendix A.1.

Upon assembling these representative tasks, we establish a task pool and proceed with the transformation process detailed in Equation 1. Ideally, executing this transformation involves human effort to refine queries, conduct searches for relevant information, and annotate the refined responses. Although this manual approach ensures quality, it is notably resource-intensive, time-consuming, and challenging to replicate. To mitigate these limitations, we employ the advanced capabilities of ChatGPT for automating the annotation process. For an in-depth description of this process and the prompts used, see Appendix A.2.

As depicted in Figure 2, our automated annotation workflow comprises several key stages, we detail it step by step as follows:

1. Begin by classifying tasks in the collected pool into the three previously mentioned categories. This step is straightforward, as each dataset corresponds to a specific data type.

2. For each dataset type, we initially use ChatGPT with a pre-defined prompt template to generate a refined query. We then employ this query to search for information from external data sources. We primarily use DuckDuckGo for most cases and treat the retrieval process as a black box.

3. Afterwards, we prompt ChatGPT to generate a renewed response based on the refined queries and its corresponding contexts. By repeating this process, we have amassed a total of about 40k instances.

## 2.2 Generator Training

After we annotate the training corpus, we can use it to train an LLM in a standard auto-regressive manner, which the objective is shown as the Equation 2.

$$\mathcal{L} = \max_{M} \mathbb{E}_{(x,y) \sim D} \left[ \log p_M(y | q_1, d_1, \ldots, q_i, d_i, x) \right] \tag{2}$$

where $\mathcal{L}$ represents the likelihood we are trying to maximize. M denotes the model parameters. The expectation $\mathbb{E}_{(x,y) \sim D}$ averages over our dataset D. $p_M(y | q_1, d_1, \ldots, q_i, d_i, x)$ is the probability of the model M generating the response y given the input x and the refined query $q_i$ with the retrieved document $d_i$ at step $i$.

## 2.3 Sampling Strategies

In this section, we introduce the inference-time strategy. As is shown in Appendix Figure 8, at each time step, our model can choose to either rewrite, decompose or disambiguate a given query, as well as opt to directly generate the response. Given this inner nature, we design a tree decoding strategy with respect to different query refinement method. However, using different queries to search yields different retrieved contexts which leads to different

final answers. That is, how to sample the most appropriate path among these trajectories is a critical part in our system. Hence, we proposed three different kind of sampling strategies, illustrated as follows:

We use $p_M$ to denote a language model with parameters $M$, and $[R_1, R_2, \ldots, R_n]$ to denote n trajectories where each trajectories contains a list of sequences which is denoted as $[X, Y]$. Here, $X$ is the input prompt, and $Y$ is the concatenation of the i intermediate steps of $Z_1, \ldots, Z_i$ (each $Z_i$ is the combination of queries and retrieved contexts) and final answer $Y_{\text{final}}$.

**PPL Based Selection**: we select the trajectory $R_{\text{final}}$ that has the lowest perplexity (PPL) on the total generated output, that is, $R_{\text{final}} = \underset{R_j \in \{R_1, \ldots, R_n\}}{\arg\min} \text{PPL}(R_j)$, where $\text{PPL}(R) = \exp\left(-\frac{1}{L} \sum_{t=1}^{L} \log p_M(Y_t | X, Y_{<t})\right)$, here $L$ is the total length of model output.

**Confidence Based Selection**: we select the trajectory $R_{\text{final}}$ that has the highest confidence on the final answer $Y_{\text{final}}$ (distinguishing it from the PPL based selection, which evaluates the total generated output), that is, $R_{\text{final}} = \underset{R_j \in \{R_1, \ldots, R_n\}}{\arg\max} \text{Conf}(R_j)$, where $\text{Conf}(R) = \sum_{t=l}^{L} \log p_M(Y_t | X, Z_1, \ldots, Z_i, Y_{<t})$, here, $t$ starts from $l$ which is the start position of the final answer, $Y_{\text{final}}$.

**Ensemble Based Selection**: We ensemble the final results by selecting the final results that has the highest cumulative confidence score, that is $Y_{\text{final}} = \underset{y}{\arg\max} \sum_{i:Y_i=y} \text{Conf}(Y_i)$.

**Upper Bound**: We also define the upper bound of our system, which means that if any one trajectory leads to the correct answer, then we consider it correct.

Our sampling methods draw inspiration from Wang et al. (2022); Yao et al. (2024), yet differ significantly in two key areas. Firstly, unlike these studies, we do not employ a larger model to assess the quality of generated trajectories; we instead utilize metrics inherent to our trained generator. Secondly, while the self-consistency approach proposed by (Wang et al., 2022) is limited to scenarios where the final answer belongs to a fixed set of options, our method is free from such constraints. See Figure 8 for clarification.

## 3 Experiments

### 3.1 Evaluation Tasks

We assess our method's effectiveness across two primary categories of Question Answering (QA) tasks: single hop and multi hop QA tasks. See Appendix B.3 for details.

**Single-Hop QA** include Arc-Challenge (Clark et al., 2018), PopQA (Mallen et al., 2022) and OpenbookQA (Mihaylov et al., 2018).

**Multi-Hop QA** include HotpotQA (Yang et al., 2018), 2WikiMultiHopQA (Ho et al., 2020) and Musique (Trivedi et al., 2022).

### 3.2 Baselines

We compare our method against a diverse set of baselines, categorized into two main groups: **No Retrieval Baselines**, which answer questions without using contexts retrieved from external databases, and **With Retrieval Baselines**, which first retrieve relevant contexts from external sources before answering based on these contexts. In both settings, our comparisons include **Llama2-7B** and **Llama2-7B-Chat**(Touvron et al., 2023) in a zero-shot configuration, as well as these models fine-tuned on **Task-Specific Datasets** (e.g., ARC_C and OBQA for single-hop QA tasks, noting that POPQA lacks a training set, thereby precluding training on this dataset and HOTPOTQA, 2WIKI, and MUSIQUE for multi-hop QA tasks) and **Our Crafted Dataset** (without intermediate steps) as a more robust baseline. Moreover, we assess against **SAIL-7B**(Luo et al., 2023) and the previously established state-of-the-art **Self-**

**RAG-7B**(Asai et al., 2024) across three single-hop QA datasets. For the multi-hop QA tasks, we also extend our method to train a **Llama3-8B** (Meta, 2024) model. The comparisons are also extended to include **Chain-of-Thought** (Wei et al., 2022), **Chain-of-Note**(Yu et al., 2023), and **Self-Ask**(Press et al., 2022), all of which are well-known methods for tasks requiring multi-step reasoning. We utilize both ChatGPT and GPT-4 as the underlying language models. We made minor modifications to the prompts to suit our tasks. For details of the prompts used, see the Appendix C.

# 4    Results and Analysis

Unless otherwise stated, our analysis is conducted using our methodology applied to Llama2-7B.

## 4.1    RQ-RAG outperforms Self-RAG and SAIL on single hop QA tasks

In Table 1, we demonstrate that RQ-RAG (ours) significantly outperforms various baseline models in both retrieval and non-retrieval settings. Specifically, under retrieval settings, RQ-RAG surpasses LLama2-7B (Zero Shot) by an average of 33.5%, highlighting the challenges LLMs face in processing retrieved contexts without search-augmented instruction tuning. Furthermore, we compare RQ-RAG against two robust baselines: 1) models supervised on specific tasks (ARC_C, OBQA) and 2) models supervised on our curated dataset but without the search-augmented intermediate step. Our findings indicate significant performance gains from supervision compared to zero-shot approaches, highlighting their competitive edge. Crucially, RQ-RAG further surpasses these baselines, demonstrating the added value of integrating a search-augmented step during training.

Additionally, we juxtapose RQ-RAG with previously established supervised, search-augmented approaches, namely Self-RAG and SAIL. Notably, our method outshines SAIL-7B by 20.3% on average across three QA tasks. Moreover, even with only about 40k training data, our method also surpasses the former state-of-the-art (Self-RAG, which utilizes 150k supervised training data) by 1.9% on average across three QA tasks.

Overall, RQ-RAG demonstrates robust performance across all evaluated tasks, firmly establishing its superiority over the aforementioned baselines.

Table 1: Performance comparison on single-hop QA tasks. The score with an arrow in parentheses indicates the comparison with the highest score within the same group.

| Model | ARC_C | POPQA | OBQA | AVG. |
|---|---|---|---|---|
| *Baseline without Retrieval* | | | | |
| **SAIL-7B** | 47.7 | 22.8 | 49.2 | 39.9 |
| **LLama2-7B** | | | | |
| + Zero Shot | 29.8 | 12.9 | 34.6 | 25.8 |
| + Zero Shot (Chat Version) | 59.9 | 14.1 | 57.6 | 43.9 |
| + SFT on Single Hop QA | 61.3 | — | 49.6 | — |
| + SFT on No Augmented Set | 62.0 | 20.5 | 64.0 | 48.8 |
| *Baseline with Retrieval* | | | | |
| **SAIL-7B** | 48.4 | 44.0 | 52.0 | 48.1 |
| **Self-RAG-7B** | 67.4 | 55.3 | 76.4 | 66.4 |
| **LLama2-7B** | | | | |
| + Zero Shot | 28.7 | 39.8 | 36.2 | 34.9 |
| + Zero Shot (Chat Version) | 53.8 | 26.4 | 40.8 | 40.3 |
| + SFT on Single Hop QA | 52.1 | — | 48.5 | — |
| + SFT on No Augmented Set | 51.5 | 45.0 | 50.2 | 48.9 |
| + RQ-RAG (Ours) | **68.3** (0.9↑) | **57.1** (1.8↑) | **79.4** (3.0↑) | **68.3** (1.9↑) |

Table 2: Performace comparison on multi-hop QA tasks.The score with an arrow in parentheses indicates the comparison with the highest score within the same group.

| Model | HOTPOTQA | 2WIKI | MUSIQUE | AVG. |
|---|---|---|---|---|
| Proprietary LLM (with Retrieval) | | | | |
| **GPT-3.5-TURBO** | | | | |
| + Chain-of-Thought | 58.6 | 43.9 | 32.3 | 44.9 |
| + Chain-of-Note | 52.5 | 34.1 | 24.6 | 37.1 |
| + Self-Ask | 62.5 | 57.0 | 37.2 | 52.2 |
| **GPT-4** | | | | |
| + Chain-of-Thought | 71.4 | 70.1 | 50.3 | 63.9 |
| + Chain-of-Note | 72.4 | 58.3 | 44.1 | 58.3 |
| + Self-Ask | 63.3 | 66.6 | 44.0 | 58.0 |
| Baseline without Retrieval | | | | |
| **LLama2-7B** | | | | |
| + Zero Shot | 6.6 | 16.0 | 3.0 | 8.5 |
| + Zero shot (Chat Version) | 3.6 | 7.9 | 1.8 | 4.4 |
| + SFT on Multi Hop QA | 34.7 | 34.2 | 6.8 | 25.2 |
| + SFT on No Augmented Set | 35.6 | 30.8 | 6.7 | 24.4 |
| Baseline with Retrieval | | | | |
| **LLama2-7B** | | | | |
| + Zero Shot | 16.7 | 18.7 | 7.4 | 14.3 |
| + Zero shot (Chat Version) | 2.8 | 3.5 | 1.8 | 2.7 |
| + SFT on Multi Hop QA | 37.5 | 32.3 | 7.9 | 25.9 |
| + SFT on No Augmented Set | 43.5 | 28.8 | 9.1 | 27.1 |
| + Self-Ask | 30.8 | 20.9 | 13.5 | 21.7 |
| + RQ-RAG (Ours) | **62.6** (19.1↑) | **44.8** (16.0↑) | **41.7** (32.6↑) | **49.7** (22.6↑) |
| **LLama3-8B** | | | | |
| + Zero Shot (Instruct Version) | 39.7 | 23.8 | 15.2 | 26.2 |
| + SFT on No Augmented Set | 71.1 | 61.0 | 40.9 | 57.7 |
| + Self-Ask | 44.5 | 44.0 | 25.9 | 38.1 |
| + RQ-RAG (Ours) | **74.2** (3.1↑) | 59.3 (1.7↓) | **48.1** (7.2↑) | **60.5** (2.8↑) |

## 4.2 RQ-RAG shows superior performance on multi-hop QA datasets

In Table 2, we present RQ-RAG's performance across three multi-hop QA datasets, reflecting trends similar to those observed in single-hop QA scenarios. When RQ-RAG is trained on either specific datasets or our curated dataset(without search-augmented step), its performance significantly surpasses that of its zero-shot counterpart. However, these baselines does not endow the model with the capability to decompose queries, a crucial skill in multi-hop scenarios where direct use of the original query for information retrieval often falls short. In contrast, our pipeline enables the model to autonomously refine queries, yielding an average enhancement of 22.6% across the three multi-hop QA datasets. Furthermore, when compared to more robust baselines that employ ChatGPT as the backbone model for answering questions, RQ-RAG significantly outperforms both the Chain-of-Thought and Chain-of-Note methods. This achievement is particularly notable given that our backbone model is considerably smaller than ChatGPT, highlighting RQ-RAG's exceptional effectiveness.

## 4.3 RQ-RAG shows high upper bound of the system

In Tables 1 and 2, we present ensemble-based selection results. For a comprehensive study of sampling strategies, we show the comparison in Figure3. Basically, we find that for single-

hop QA tasks, the confidence-based strategy generally excels, while for multi-hop QA tasks, the ensemble-based strategy shows superior performance. Overall, the ppl-based strategy demonstrates moderate performance in comparison. Beyond these strategies, we also assess the upper bound of our system by considering the success rate across all trajectories; success is defined as any trajectory that leads to the correct answer. This comprehensive evaluation reveals high potential upper bound for our system. Specifically, it reaches 76.8% in ARC_C, 65.6% in POPQA, 84.0% in OBQA, 80.5% in HOTPOTQA, 60.6% in 2WIKI, and 54.5% in MUSIQUE, respectively. These results underscore the system's ability to refine queries and retrieve varying contexts multiple times, thereby increasing the likelihood of reaching the correct answer. An avenue for future enhancement involves devising more effective methods to select among generated trajectories, such as leveraging LLMs for scoring each one. Furthermore, it's important to note that the current upper bound does not represent the system's absolute limit; additional improvements could be as well achieved through context reranking and explicit noise reduction in retrieved contexts.

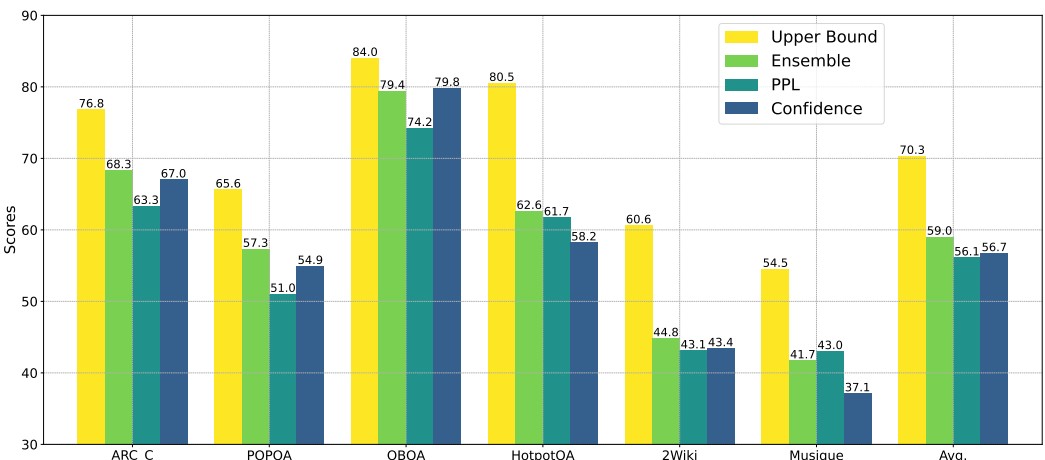

Figure 3: Performance of different sampling strategies on six tasks.

## 4.4   Regenerating answers grounded on contexts matters

As mentioned in Section 1, a key innovation of our method compared to previous approaches is the use of ChatGPT to regenerate answers based on provided contexts, rather than relying on original answers from the dataset. In this section, we explore the impact of varying the proportion of the original answer retained from the dataset. To ensure a fair comparison, we maintain consistent training hyper-parameters across experiments. We curated our search-augmented dataset by retaining 0% (our primary approach), 25%, 50%, 75%, and 100% of the original answer, then assessed the effects on performance across three multi-hop QA datasets. Figure 4a illustrates our findings on HotpotQA, with the all-regenerated answer setting (0% retention) showing the best performance. Additionally, there is a noticeable decline in effectiveness as the proportion of the original answer retained increases, indicating that our strategy of regenerating answers based on retrieved contexts is beneficial. We show the results of the other two datasets in the Appendix D.

## 4.5   A closer look into the effect of different action on query refinement.

In this section, we aim to illustrate the effect of each different action on query refinement. Our methodology employs sampling strategies to select the best trajectories; however, we want to evaluate the effectiveness of deliberately using specific action for all instances on three tasks. As shown in Figure 4b, we compare the use of a single action during inference with our designed sampling strategies (as the one used in Table 2). The observations are: 1) no single action outperforms the others across all three tasks; 2) while some actions yield

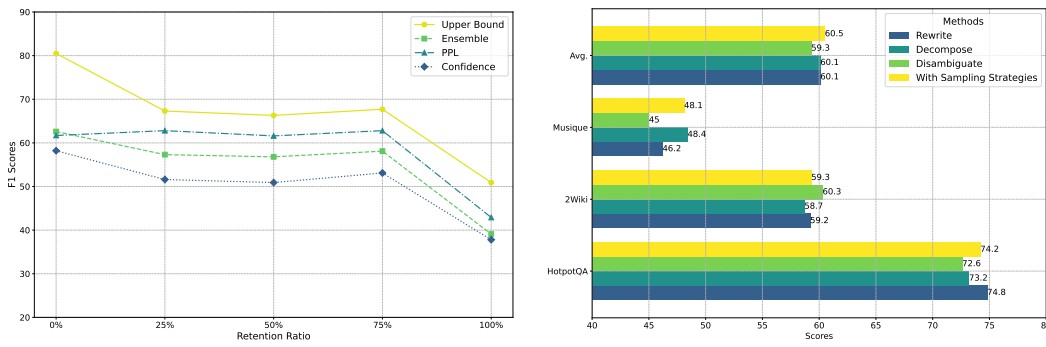

(a) The impact of retention ratio on HotpotQA.      (b) The impact of each single action.

Figure 4: **Left**: Increasing the ratio of the original answers in the dataset results in a decreased F1 score. **Right**: No single action outperforms the others across all three tasks.

better results for a specific task compared to the sampling strategies, they do not perform better overall, indicating the benefit of our designed sampling strategies.

Furthermore, we investigate the proportion of each action selected by our strategies. Figure 5 demonstrates the result for HOTPOTQA, showing that although the disambiguation process accounts only for 10% of our training data, the proportion of different actions selected by the model is almost balanced. For the results of the other two tasks, see Figure 11 and Figure 12. The results in this section are based on Llama3-8B.

### 4.6 Our system is resilient to different data resources

In this section, we investigate the impact of utilizing various data sources. Our dataset primarily leverages DuckDuckGo for data retrieval during dataset curation, yet our findings suggest that the choice of data sources during inference has a minimal effect on our system's performance. This experiment evaluates our system across three single-hop QA tasks, employing DuckDuckGo, Wikipedia, and Bing Search as the data sources. Table 3 demonstrates that our system exhibits greater resilience to changes in data sources compared to Self-RAG, which primarily uses Wikipedia for data retrieval during dataset curation. Notably, using Bing Search results in a performance drop of 3-5% for Self-RAG across the tasks. In contrast, our method, curated using DuckDuckGo, shows negligible impact on performance (a variance of 0.7 compared to their 1.8) when different data sources are used during inference.

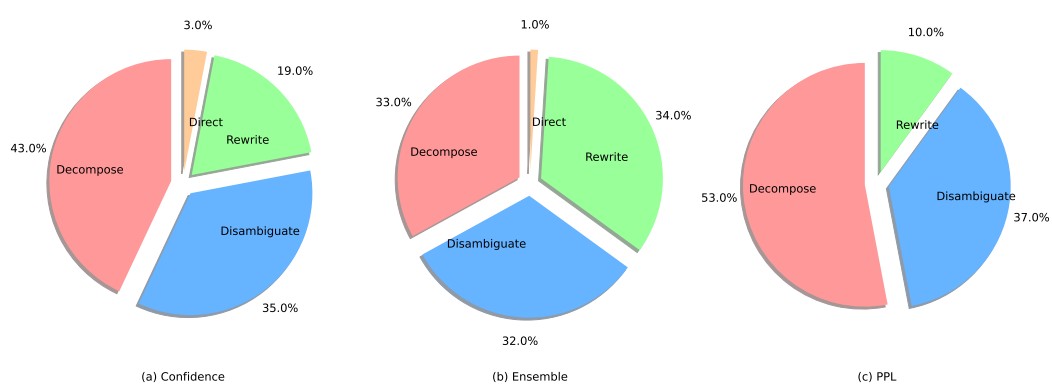

Figure 5: The ratio of all the action selected in different sampling strategies on HOTPOTQA.

| Method | ARC_C | POPQA | OBQA | AVG. ↑ | VAR. ↓ |
|---|---|---|---|---|---|
| Self-RAG | | | | | |
| + DuckDuckGo | 67.4 | 55.3 | 76.4 | | |
| + WIKI | 67.3 | 54.9 | 78.0 | 65.5 | 1.8 |
| + BingSearch | 64.6 | 49.0 | 76.8 | | |
| Ours | | | | | |
| + DuckDuckGo | 68.3 | 57.1 | 79.8 | | |
| + WIKI | 67.8 | 52.6 | 80.6 | **67.6** | **0.7** |
| + BingSearch | 67.9 | 55.6 | 78.8 | | |

Table 3: Performance comparison of different retrieval sources.

## 5 Related Works

Retrieval-augmented generation (RAG) represents an effective strategy for enhancing a model's capability to leverage non-parametric knowledge by retrieving external data resources, rather than relying solely on its intrinsic parametric knowledge during generation. This paradigm has garnered widespread attention across both industry and academia, proving its efficacy in a variety of scenarios such as question answering (Lewis et al., 2020), code generation (Zhou et al., 2022), alignment with human values (Xu et al., 2023b) and reducing hallucinations (Shuster et al., 2021). Recent advancements in RAG systems can generally be categorized into two areas: improvements in either the retrieval component or the generation component of the system.

**Retriever in Retrieval Augmented Generation**

Previous studies have underscored the critical role of the retrieval component in RAG systems, illustrating how LLMs can be susceptible to irrelevant contexts (Shi et al., 2023a). Systems that retrieve more semantically relevant contexts (Karpukhin et al., 2020) significantly outperform those based on BM25 (Robertson et al., 2009), demonstrating a substantial improvement. More recently, certain works have highlighted that LLMs themselves can act as a supervisory signal for training the retrieval component (Shi et al., 2023b), or even serve as the retrieval component (Sun et al., 2022), thanks to their extensive knowledge and robust capabilities. Another research direction involves denoising irrelevant context by employing additional models to explicitly filter out (Yoran et al., 2023) or compress (Xu et al., 2023c) such contexts. In our work, we approach the retrieval component as a black box, directly using the contexts returned from search engine APIs or a pre-provided data corpus. This method is compatible with denoising techniques, which could further enhance our system's performance; exploring these techniques will be an avenue for future research.

**Generator in Retrieval Augmented Generation**

In contrast to efforts aimed at enhancing the retrieval component of RAG systems, another research focuses on optimizing the generator part. SAIL (Luo et al., 2023) trains the LLM to differentiate irrelevant contexts during the generation process. Self-RAG (Asai et al., 2024) trains the LLM to self-reflective on the retrieved contexts. Rewrite-Retrieve-Read (Ma et al., 2023) trains a small model to rewrite queries for a black-box reader while we train the LLM to refine queries by itself in this paper.

## 6 Conclusion

In this paper, we introduce RQ-RAG, a framework that enhances LLMs by training them on a meticulously curated dataset. This training enables the LLMs to refine queries through a process of rewriting, decomposing, and disambiguating. Our experimental findings demonstrate that RQ-RAG not only surpasses previously established SOTA method across three single-hop QA tasks but also exhibits superior performance in complex multi-hop QA scenarios, even when compared to the exceptional ChatGPT, underscoring great effectiveness of our approach.

**Acknowledgments**

This work is funded in part by the Theme-based Research Scheme grant(No.T45-205/21-N) and the InnoHK funding, Hong Kong SAR.

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

## A  Data Collection

### A.1  Data Statistics

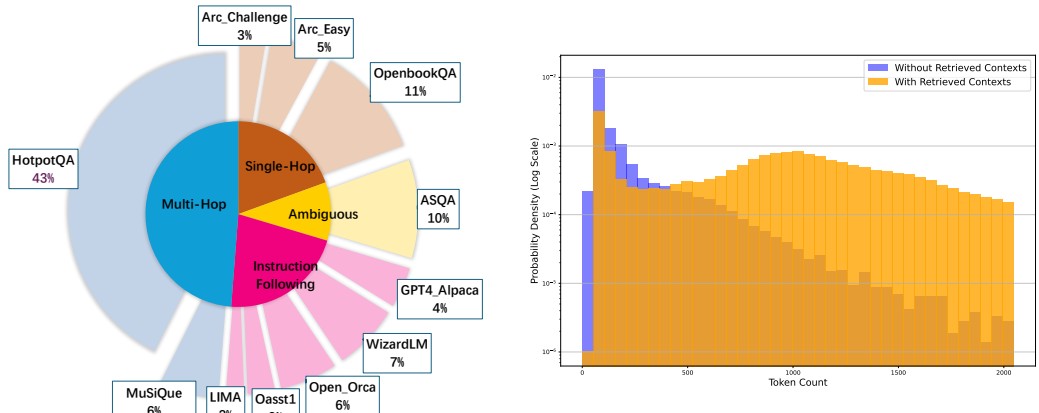

Figure 6: Data categories.

Figure 7: Token Distribution.

Figure 6 shows the categories we include in our data curation process, and the specifc dataset we used. Primarily, our data contains single-hop QA tasks including Arc-Easy/Arc-Challenge (Clark et al., 2018), OpenbookQA (Mihaylov et al., 2018), multi-hop QA tasks including HotpotQA (Yang et al., 2018), Musique (Trivedi et al., 2022), ambiguous tasks including ASQA (Stelmakh et al., 2022), and in order to equip our model with general capabilities, we also add instruction following tasks including LIMA (Zhou et al., 2024), WizardLM (Xu et al., 2023a), Open-Orca (Mukherjee et al., 2023), OpenAssistant (Köpf et al., 2024), GPT4-Alpaca (Taori et al., 2023). In total, we collect 42810 instances.

Moreover, as illustrated in Figure 7, the distribution of tokens by probability density is significantly altered by search augmentation. Our augmented dataset, exhibits a considerable increase in token count, with the majority of instances spanning from 150 to 2000 tokens. In contrast, the original dataset presents a notably shorter length, with most entries falling under 200 tokens.

### A.2  Data Annotation

In our study, we employed ChatGPT, specifically the **gpt-3.5-turbo-0125** version, for data annotation, setting the temperature parameter to 0 to enhance reproducibility. Throughout the annotation process, we encountered certain limitations with ChatGPT, including its occasional refusal to answer and failure to adhere to the specified output format. Instances exhibiting these issues were subsequently excluded.

Table 4, Table 5 and Table 6 display the prompt templates we used to interact with ChatGPT across different phases of annotation process, as detailed in Section 2. In these templates, the blue text serves as a placeholder for specific input.

## B  Experimental Details

### B.1  Training Hyperparameters

In this paper, all of the models are trained on 8 NVIDIA H800 with 80GB memory. And all models are trained 1 epoch using learning rate of 2e-5 with 3% warmup steps. Given the extended context length of our dataset, we set the maximum input length at 4096.

In a multi-turn dialogue scenario, your task is to determine whether it is necessary to use a search engine to answer a user's query and provide a list of possible search queries.

Consider the following two scenarios:

Non-Informational Replies: Sometimes, users may respond with statements or expressions that do not require information retrieval, such as "thank you" or "okay." In these cases, assess whether a search engine query is necessary.

Ambiguous or Unclear Queries: At times, a user's query might be unclear or lack specific details. Your role is to recognize the user's intent and rewrite the query to make it clearer and more precise, facilitating an effective search engine query.

Previously Answered Queries: Check if the current query or a similar one has been previously asked and answered in the conversation history. If relevant information or evidences have already been provided, acknowledge this and avoid repeating the search.

Based on the above 3 scenarios, please reply with the following format strictly:

For the case that do not need to query the search engine, output as follows:

{In context examples}

—

As outlined, it is necessary to output Retrieval Necessity first, and the output should be one of the "yes" and "no" and the query for the search engine should be split by a line break. For most of the case, retrieval process might help you better answer the question, only skip the retrieval process when you are fairly confident about not doing so.

Now, please answer:

Conversation History:

{Conversation History}

Current User's Query:

{Current User's Query}

Response For Retrieval Necessity:

Table 4: Multi-Turn dialogue prompt template during data construction.

Your task is to effectively decompose complex, multihop questions into simpler, manageable sub-questions or tasks. This process involves breaking down a question that requires information from multiple sources or steps into smaller, more direct questions that can be answered individually.

Here's how you should approach this:

Analyze the Question: Carefully read the multihop question to understand its different components. Identify what specific pieces of information are needed to answer the main question.

Here are an example of how you should solve the task:

{In context examples}

—

As outlined, please format your answer as multiple lines of text. Ensure that each subsequent question follows from the previous one and is self-contained and be capable of being answered on its own. Ensure there is exactly one line break between each line.

Now please answer:

Provided Contexts:

{Provided Contexts}

Multihop Question:

{Multihop Question}

Decomposed queries:

Table 5: Query decomposition prompt template during data construction.

> Your task is to identify and resolve ambiguity in complex questions, ensuring they are clear and unambiguous. This requires pinpointing elements of the question that could be interpreted in more than one way and refining the question to ensure a single, clear interpretation.
> Approach this task as follows:
> Analyze the Question: Read the question thoroughly to identify ambiguous parts. Consider the different ways the question could be interpreted based on its current wording.
> Clarify the Query: Reformulate the question to eliminate ambiguity. This may involve specifying details, narrowing down broad terms, or providing additional context to guide the interpretation.
> Here's an example of how to complete the task:
> For example:
> {In context examples}
> —
> As outlined, please format your answer as multiple lines of text. Ensure there is exactly one line break between each line.
> Now, please answer:
> Original Question:
> {Original Question}
> Disambiguated Query:

Table 6: Query disambiguation prompt template during data construction.

## B.2 Sampling Strategies

Figure 8 depicts our tree decoding strategy through a structured flow. This strategy unfolds by controlling expansion paths via special tokens, generating and retrieving queries iteratively: a process of generate → retrieve → generate → retrieve → . . . → answer. At each iteration, the model decodes diverse search queries tailored to specific needs—whether to rewrite, decompose, or disambiguate. These queries, in turn, fetch distinct contexts, leading to varied expansion paths. Thus, within predetermined exploration width and depth, our approach facilitates the generation of multiple trajectories. Given this diversity, it becomes imperative to adopt a method for accurately sampling the optimal trajectory, as discussed in Section 2. Experimentally, we set the exploration depth to 2 for three single-hop QA tasks and adjusted it to 2 for HotpotQA and 4 for 2WikiMultihopQA and MuSiQue, respectively.

## B.3 Evaluation Datasets and Metric

**Single-Hop QA**:

For the Arc-Challenge (Mallen et al., 2022), comprising 1172 four-choice QA instances, we measure model performance using accuracy. In the case of PopQA (Mihaylov et al., 2018), we focus on a longtail subset of 1399 instances and adopt a match score metric to assess if the model's output includes the ground truths. For OpenbookQA (Mihaylov et al., 2018), comprising 500 four-choice QA instances, we measure model performance using accuracy.

**Multi-Hop QA**:

We evaluate multi-hop QA on a sample of 500 instances following Trivedi et al. (2023). Our experiments utilize datasets from HotpotQA-distractor (Yang et al., 2018), 2Wiki-MultihopQA (Ho et al., 2020), and MuSiQue-Ans (Trivedi et al., 2022) within a reading comprehension setting, where the candidate documents are from their original datasets. For HotpotQA and 2WikiMultihopQA, each question is linked to 10 passages, of which only a few (2 for HotpotQA and 2 or 4 for 2WikiMultihopQA) are relevant. MuSiQue-Ans presents a greater challenge, offering 20 candidate documents and featuring questions that require 2, 3, or 4 hops to answer. We employ the F1 score as our primary performance metric.

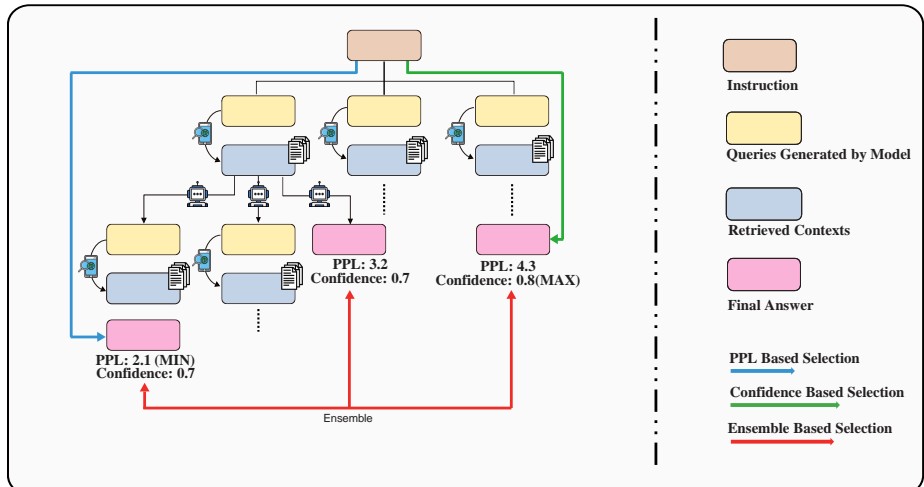

Figure 8: Given different paths, we develop three different strategies, ppl based, confidence based and ensemble based selection.

### B.4   Retriever Setup

During data curation, we primarily employ DuckDuckGo to gather relevant contexts for most scenarios, selecting the top-3 documents. Each result includes a title, a preview snippet, and the webpage's URL. To simplify our process, we utilize only the title and preview text without further scraping the url. For multi-hop QA tasks, we leverage BM25 to extract contexts from candidate documents provided by specific datasets, discarding instances where the contexts do not align with the support documents.

During inference for single-hop QA tasks, our standard approach involves retrieving three contexts from DuckDuckGo. For multi-hop QA scenarios, we utilize the **text-embedding-3-large** model (OpenAI, 2023) to identify three most relevant contexts from the candidate documents at each step.

## C   Prompt for COT, CON and Self-Ask

See Figure 7, Figure 8, Figure 9 for details.

## D   Additional Results

We show the impact of retention ratio on another two multi-hop QA tasks in this section. As is discussed in Section 4.4, similar phenomena are observed in Figures 9 and 10, indicating that regenerating answers grounded on the provided contexts is crucial which will in turn affect the performance of downstream tasks . Additionally, the ratio of all the actions selected in different sampling strategies on two other multi-hop QA tasks is shown in Figure 11 and Figure 12. Overall, a similar trend can be observed where the direct answer action is not selected by PPL strategies in any of the three tasks. Moreover, the ratio of direct answer actions is minimal across all strategies and tasks. Generally, the three different actions are balanced across all tasks and strategies.

Task Description:
You are given three questions: Question Q1, Question Q2, and Question Q3, along with the passages P1 for Q1, and the passages P2 for Q2. The requirements are as follows:
1. Please answer Q1 and Q2 before answering Q3.
2. For Q3, give the final answer in the format 'The answer is _'. Ensure the final answer is short and concise. Do not repeat the question in the final answer.
Q1:
{Q1}
P1:
{P1}
Q2:
{Q2}
P2:
{P2}
Q3:
{Q3}
Let's think step by step:

Table 7: COT prompt template that we use for our baseline.

Task Description:
You are given three questions: Question Q1, Question Q2, and Question Q3, along with the passages P1 for Q1, and the passages P2 for Q2. The requirements are as follows:
1. For Q1 and Q2, read the given question and the passages to gather relevant information.
2. For Q1 and Q2, write reading notes summarizing the key points from these passages.
3. For Q1 and Q2, discuss the relevance of the given question and context.
4. For Q1 and Q2, if some passages are relevant to the given question, provide a brief answer based on the context.
5. For Q1 and Q2, if no passage is relevant, directly provide an answer without considering the context.
6. For Q3, review the reading notes for Q1 and Q2, and give the final answer in the format 'The answer is _'. Ensure the final answer is short and concise. Do not repeat the question in the final answer.
Q1:
{Q1}
P1:
{P1}
Q2:
{Q2}
P2:
{P2}
Q3:
{Q3}

Table 8: CON prompt template that we use for our baseline.

Given the following question, answer it by providing follow up questions and intermediate answers. For each follow up question, you are given a context which is the top returned Wikipedia snippets for the question. If no follow up questions are necessary, answer the question directly.
#
{Knowledge}
{Knowledge}
Question: What is the name of this American musician, singer, actor, comedian, and songwriter, who worked with Modern Records and born in December 5, 1932?
Are follow up questions needed here: Yes.
Follow up: Who worked with Modern Records? Intermediate answer: Artists worked with Modern Records include Etta James, Little Richard, Joe Houston, Ike and Tina Turner and John Lee Hooker.
Follow up: Is Etta James an American musician, singer, actor, comedian, and songwriter, and was born in December 5, 1932?
Intermediate answer: Etta James was born in January 25, 1938, not December 5, 1932, so the answer is no.
Follow up: Is Little Richard an American musician, singer, actor, comedian, and songwriter, and was born in December 5, 1932?
Intermediate answer: Yes, Little Richard, born in December 5, 1932, is an American musician, singer, actor, comedian and songwriter.
So the final answer is: Little Richard
. . .
{In context examples}
—
Questions:

Table 9: Self-Ask prompt template that we use for our baseline.

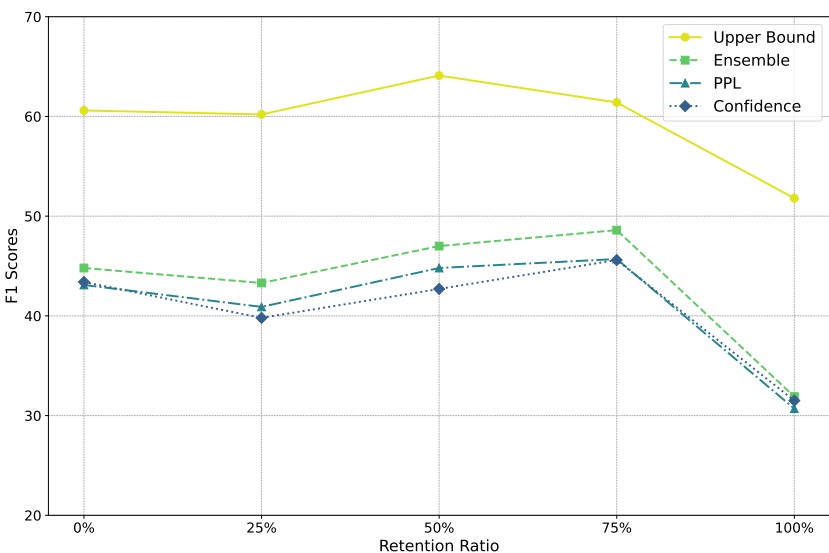

Figure 9: The impact of retention ratio on 2WikiMultihopQA.

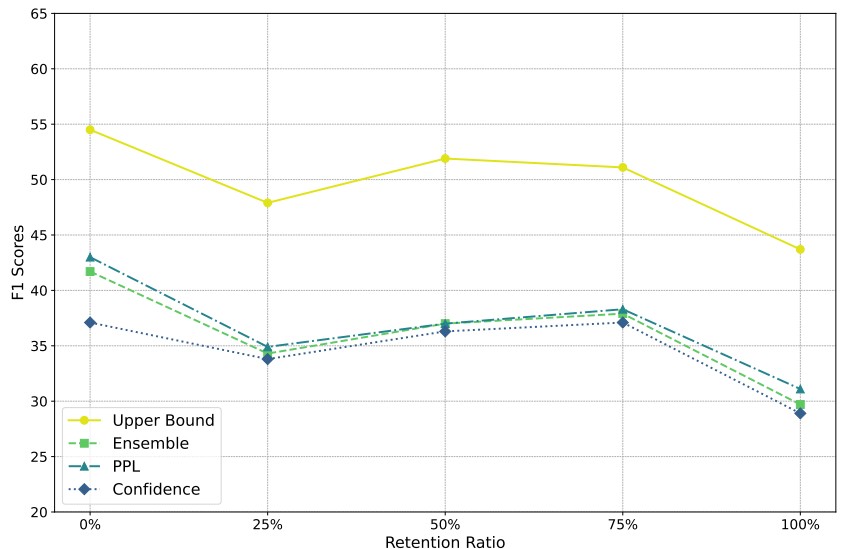

Figure 10: The impact of retention ratio on MuSiQue.

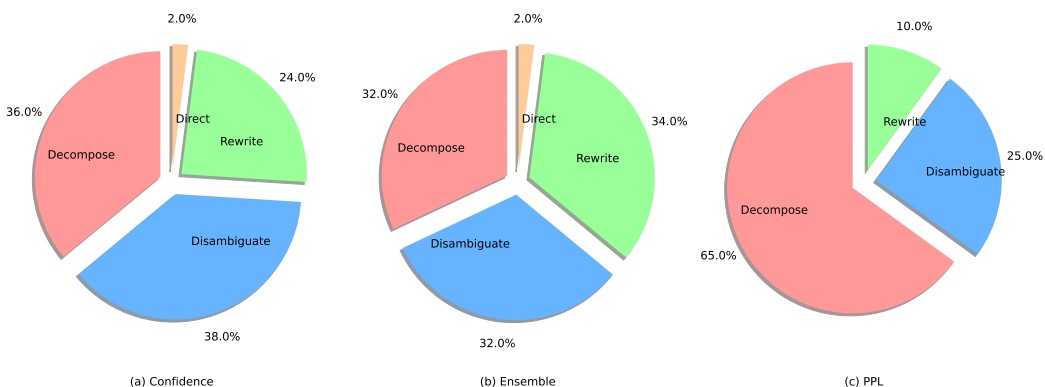

Figure 11: The ratio of all the action selected in different sampling strategies on 2WIKI.

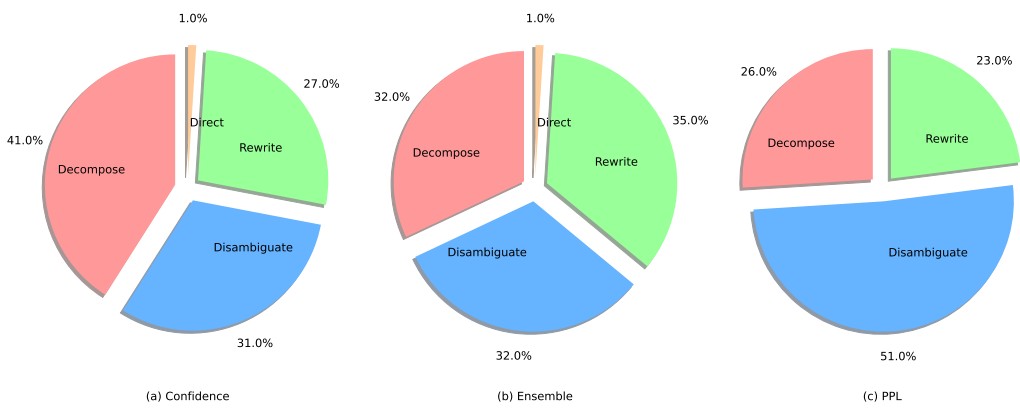

Figure 12: The ratio of all the action selected in different sampling strategies on MUSIQUE.

