# OpenReview forum: "RQ-RAG: Learning to Refine Queries for Retrieval Augmented Generation"
_colmweb.org/COLM/2024/Conference — COLM_

### Official Review · Reviewer_H1CE · 2024-04-20

**Rating:** 5
**Confidence:** 4
**Ethics Flag:** 1

**Summary:**

Traditional retrieval-augmented systems often face two main issues: (1) the irrelevant content in retrieved documents may mislead the model and (2) searching with the original query often fails to retrieve adequate information. To this end, this paper introduces a method called Refine Query for Retrieval Augmented Generation (RQ-RAG), which trains a 7B Llama2 model to adaptively rewrite, decompose, and disambiguate the original input query with their collected datasets. They conduct experiments on both single-hop and multi-hop QA datasets. The results show the effectiveness, high potential, and robustness of their method.

**Reasons To Accept:**

1. Positive experimental results: The method proposed in this paper outperforms existing state-of-the-art retrieval-augmented methods like Self-RAG-7B on multiple datasets, with detailed comparative tests against the baseline of the llama-7B model.

2. Comprehensive experiments: The paper also demonstrates the effectiveness, rationality, and robustness of its method through experiments on the proportion of dataset enhancement, sampling strategies, and the sources of retrieved information.

**Reasons To Reject:**

1. The paper proposes three methods of data enhancement: rewrite, decompose, and disambiguate. I am still concerned about how much these three methods actually contribute to performance improvement and whether they are all effective.

2. For the MQA (Multi-Hop QA) task, the paper only compares with the Llama2-7B backbone baseline, which shows significant improvements. However, in the MQA field, knowledge editing is also a widely studied method (e.g., MeLLo [1], PokeMQA [2]). I recommend the authors provide comparisons with these similarly high-performing knowledge editing methods.


3. The refined query method proposed in the paper adds Decomposition and Disambiguation to the queries. Is this effective for every query? Could this approach introduce bias in the processing of some original data?


[1] Mquake: Assessing knowledge editing in language models via multi-hop questions
[2] PokeMQA: Programmable knowledge editing for Multi-hop Question Answering

---

> ### Author Rebuttal · Authors · 2024-05-31
>
> Thanks for your comments.
>
> **1. Comparison with Knowledge Editing Methods (e.g., MQuAKE and PokeMQA)**
>
> To clarify, our work aims to enhance small models to explicitly refine queries for better answering both single-hop and multi-hop QA tasks, which do not overlap with the knowledge editing area mentioned by the reviewer.
>
> We outline the differences between our work and knowledge editing as follows;
>
> Knowledge editing addresses the issue that information stored in LLM becomes outdated quickly, and retraining from scratch is often not feasible. This has led to techniques for injecting new facts by updating model weights. Although knowledge editing and our work aim to address outdated parametric knowledge, our approach focuses on retrieval-augmented generation to improve factuality and reliability. Knowledge editing requires a set of new knowledge to be updated, focusing on effectively updating these new knowledge without disrupting other facts. In our work, no edited facts are provided, making the methods suggested by the reviewer **inapplicable**.
>
> **2. The Effect of Different Actions**
>
> As mentioned in response to reviewer PYYL, we carried out new experiments addressing the effect of different actions. For completeness, we include the results here.
>
> **Table1: The Ratio of Different Actions Under Different Sampling Strategies for 3 Tasks**
>
> |  | Decompose | Disambiguate | Rewrite | Direct |
> |----------------|-----------|--------------|---------|--------|
> | **Confidence** | 0.46      | 0.35         | 0.19    | 0.03   |
> | **Ensemble**   | 0.34      | 0.32         | 0.34    | 0.01   |
> | **PPL**        | 0.53      | 0.37         | 0.10    | 0.00   |
>
> **Table2: F1 Score for Deliberately Using a Specific Action for All Instances of 3 Tasks**
>
> |  | HotpotQA | 2Wiki | Musique | Avg   |
> |--------------------------|----------|-------|---------|-------|
> | **Rewrite**              | 74.8     | 59.2  | 46.2    | 60.1  |
> | **Decompose**            | 73.2     | 58.7  | 48.4    | 60.1  |
> | **Disambiguate**         | 72.6     | 60.3  | 45.0    | 59.3  |
> | **With Sampling Strategies** | 74.2     | 59.3  | 48.1    | 60.5  |
>
> **Main Findings:**
> - The proportion of different action selections is almost balanced, indicating minimal bias in the data as the reviewer was concerned (Table 1).
> - Simply choosing the same action might not yield the best performance. Each action contributes uniquely, and combining different actions leads to better overall performance (Table 2).

---

> > ### Author Response · Authors · 2024-06-03
> > **Follow-Up on Rebuttal Response**
> >
> > Dear Reviewer,
> >
> > Thank you for your comments on our paper. I believe that our rebuttal addresses the issues you raised, and the newly provided experimental results further substantiate the novelty of our work. If you have any additional questions or concerns, please feel free to reach out.
> >
> > Thank you once again for your time and consideration.

---

### Official Review · Reviewer_KkQG · 2024-05-10

**Rating:** 6
**Confidence:** 5
**Ethics Flag:** 1

**Summary:**

This paper proposes an approach similar to Self-RAG or Self-Ask with specific actions to do multiple steps of retrieval and rewrite queries as needed. This system-like approach is pretty effective on various single-hop and multi-hop QA tasks. Notably, it outperforms Self-RAG on single-hop QA, although the baselines for multihop-QA do not seem particularly strong.

The main weakness is the writing is not very clear on the motivations for these actions, nor how their approach is specifically different to previous methods. For example, self-ask is not mentioned as a baseline and already includes multi-step retrieval and question decomposition.

**Questions To Authors:**

Why not try other self-consistency alternatives, such as universal SC?

How is PPL different from beam search?

**Reasons To Accept:**

1. The results compared to self-RAG are compelling. Although the multi-hop results seem to be missing key baselines.
2. The ablations are informative for choosing between search engines, although it would be nice to see more discussion about retrieval accuracy across different search engines, such as whether they exhibit similar behavior and prefer different kinds of queries.
3. The method is simple and flexible, depending mostly on synthetic labels. That being said, more could have been discussed about the dataset construction in the main text.

**Reasons To Reject:**

1. The paper is not self-contained. Crucial details for the dataset construction are missing or left to the appendix. Details on baselines are not included at all, e.g. how does this differ from Self-RAG?
2. The llama-2-7b baselines for multi-hop QA are not very effective, and it's a bit unclear whether this is because they do not conform to the task well, or they have some lack of capability.
3. The self-ask approach is not discussed, yet it is a well known method for tasks like multi-hop QA with search engines.
4. The Llama-2-7b is a weak model compared to various other models that would have been available at the time, accessible through APIs such as together.ai. For example, llama-2-70b. It is not clear whether their method will still be effective for these stronger options.

To make it more clear, the main weakness are the lack of self-ask baseline (and potentially other well known baselines) and general lack of details in the main text. Including those types of details would certainly strengthen the paper, although it may reveal limited novelty or effectiveness of the presented approach.

---

> ### Author Rebuttal · Authors · 2024-05-31
>
> We appreciate your constructive feedback and would like to address your concerns as follows:
>
> **1. Difference with Self-Ask**
>
> The reviewers mentioned that a primary weakness is the distinction between our approach and self-ask. Here, we justify the novelty of our work.
>
> First, self-ask relies on large language models with strong instruction-following abilities. It barely works with smaller models that cannot refine their answers without further training. Our method meticulously curates a retrieval-augmented instruction tuning dataset, training a small model to possess the ability to refine queries.
>
> Second, self-ask performs multi-step reasoning implicitly, restricting users' control over the types of follow-up queries. Conversely, our method uses special tokens to explicitly guide the model to generate various types of query refinements, providing greater flexibility.
>
> Additionally, we provide new results for the methods for completeness as follows;
>
> |              | HotpotQA | 2Wiki | Musique | Avg   |
> |---------------------|----------|-------|---------|-------|
> | **LLaMA3 8B (RQ-RAG)**| 74.2     | 59.3  | 48.1    | **60.5**  |
> | **Self-Ask**        |          |       |         |       |
> | **LLaMA2 7B**       | 30.8     | 20.9  | 13.5    | 21.7  |
> | **LLaMA3 8B**       | 44.5     | 44.0  | 25.9    | 38.1  |
> | **GPT-3.5**         | 62.5     | 57.0  | 37.2    | 52.2  |
> | **GPT-4**           | 63.3     | 66.6  | 44.0    | 58.0  |
>
> ***Highlight***
>
> Our newly trained model on LLaMA3 8B surpasses the self-ask method on both GPT-3.5 and GPT-4, suggesting the strong capability of our method and its robustness effect over stronger options. We expect this response to resolve the issue of more strong baselines and solidify its effectiveness.
>
> **2. Difference Between PPL and Beam Search**
>
> In our scenario, the sampling strategies we proposed resemble beam search. At each step, the different refinement action can be seen as a node to be explored. Among these nodes, there exists the most correct one (we use the upper bound to indicate the high potential of our system). The PPL or your mentioned universe SC can be seen as the heuristic to select the top-k node to expand. Under these circumstances, various heuristics can be proposed in our system. In this work, we preliminarily try out PPL and confidence, while other heuristics can be researched to further enhance the performance of our system.

---

> > ### Author Response · Authors · 2024-06-03
> > **Follow-Up on Rebuttal Response for Paper**
> >
> > Dear Reviewer,
> >
> > Thank you for the time and effort you have put into evaluating my work. As the review process is nearing completion, I wanted to kindly inquire if there are any unresolved issues or concerns that you still have. Should you have any further questions, I am more than happy to address them.
> >
> > Thank you once again for your consideration.

---

> > ### Comment · Reviewer_KkQG · 2024-06-03
> >
> > I've improved my score to 6 based on the new results.
> >
> > The connection to beam search sounds tenuous. I am a bit more confused about the approach based on this description, since it sounds more like beam=1, but that is not guaranteed to find argmin and it is only an approximation (in general, finding argmin is a challenging search problem). This is also different from my understanding of the method as described in the paper which samples multiple trajectories rather than incrementally does beam search.

---

> > > ### Author Response · Authors · 2024-06-05
> > > **Thank you for your continued support of our paper.**
> > >
> > > Thank you for your continued support of our paper. We appreciate the opportunity to address your recent confusion regarding our methods' connection with beam search. Allow us to provide clarification.
> > >
> > > We do not do the beam=1 search, instead, we explore the node as much as we can (although we have set the width=3 in our experiment, that means at each step, the model can either generate:
> > >
> > > [rewritten query1, rewritten query2, rewritten query3] or
> > > [rewritten query1, rewritten query2, decomposed query1] or
> > > [rewritten query1, decomposed query1, decomposed query2] or
> > > ...
> > > and so on.
> > >
> > > For instance, if the maximum depth is set to 2 and maximum width is set to 3, then the total trajectories will be 9 (3^2); note that for token-level generation, we adopt greedy decoding. After we gather multiple trajectories leading to the potential answers, we use the strategies we describe in the paper (ppl, confidence, ensemble) to extract the final answer as our predicted answer.
> > >
> > > I hope the above explanation adequately addresses your question.

---

### Official Review · Reviewer_xvu9 · 2024-05-10

**Rating:** 6
**Confidence:** 4
**Ethics Flag:** 1

**Summary:**

This paper introduce a new training and inference method for retrieval-augmented generation, which enables an LM learn to retrieve when necessary and re-write compositional and ambiguous queries. The proposed method is based on recent retrieval-augmented instruction-tuning method, particularly Self-RAG (e.g., using special tokens to control model-behaviors). While in Self-RAG, the authors only train models for single-turn settings and reuse the input-output data from existing datasets, this paper re-generate output data based on retrieved documents, which authors find crucial to achieve the strong performance. Experimental results on single-hop and multi-hop queries indicate the effectiveness of proposed method.

**Reasons To Accept:**

- The proposed method is well-motivated and carefully designed.
- While there are several work that leverages existing datasets for retrieval-augmented instruction-tuning, the authors empirically explore answer regeneration strategy that regenerates outputs of existing datasets based on the retrieved answers. This may shed light on improved instruction-tuning pipeline for retrieval-augmented LMs.
- Experimental results show the effectiveness of proposed method compared with prior method such as SAIL or Self-RAG.

**Reasons To Reject:**

- Limited novelties: as also noted by the authors, the proposed method is heavily influenced by prior work such as Self-RAG or SAIL, and the novelties newly introduced by this work may be limited.
- The baseline models are somewhat weak — several recent work shows strong performance on the datasets, and also to my understanding, the ChatGPT and GPT-4 baselines do not use retrieval. Although I don’t think the proposed method should outperform proprietary models whose training details are private, claiming superiority over ChatGPT when ChatGPT baseline doesn’t use retrieval seems unfair to me.
- Evaluations are mostly done in short-form QA tasks, and it is an open question how well we can apply the proposed setup for long-form generation tasks. For instance, the ensemble method shows the best performance on most of the tasks, but ensembling long-form outputs are not trivial.

---

> ### Author Rebuttal · Authors · 2024-05-31
>
> Thanks for your insightful comments!
>
> **1. Claims of Novelty and Contributions**
>
> The main concern is how our method compares with Self-RAG and SAIL. We would like to clarify and highlight the novelty of our approach as follows:
>
> Solely relying on RAG with its non-parametric parts shows some limitations, including but not limited to:
> 1. It requires strong instruction-following ability to reply considering the provided context, due to the different format from their original sft dataset.
> 2. It lacks interpretability when the retrieved documents contradict its parametric knowledge.
>
> To address the above issues, our work focuses on effectively constructing retrieval-augmented instruction following datasets to train the model for internal learning and designing an effective mechanism for inference. This shares a common goal with previous work like self-RAG and SAIL. This methodology has also been explored by other works [1] after we submitted our research, highlighting the potential and value of this direction. While we draw inspiration from data construction processes such as self-RAG and SAIL, our key contribution lies in **how** we construct the instruction tuning dataset. This is where our paper differs from previous work and makes a significant contribution:
>
> 1. We explicitly make the model generate refined queries with more interpretability and controllability.
> 2. We do not reuse the original dataset output and claim that regenerating it is significant. To our best understanding, this is the first attempt to do so, and we view it as the novelty of this paper.
>
> [1] Modarressi, A., et al. (2024). MemLLM: Finetuning LLMs to Use An Explicit Read-Write Memory. arXiv preprint arXiv:2404.11672.
>
> **2. Fairness When Compared to ChatGPT**
>
> Sorry for causing the misunderstanding. The method we compared (chain-of-thought and chain-of-note) uses retrieval settings where we utilize the generated thoughts and notes to retrieve relevant documents. Therefore, the comparison is **FAIR**, and we will make this clearer in our next version. Thank you for pointing this out.
>
> Additionally, we have provided new results on our method trained on the new model LLaMA3 8B, which shows comparable results even to gpt4. This demonstrates the strong effectiveness and robustness of our method. Due to the limited characters allowed, please refer to the results shown in response to reviewer KkQG for more details.

---

> > ### Comment · Reviewer_xvu9 · 2024-06-05
> > **Thank you for your response!**
> >
> > Thank you for your detailed response. I'll keep my score (6).

---

### Official Review · Reviewer_PYYL · 2024-05-11

**Rating:** 7
**Confidence:** 4
**Ethics Flag:** 1

**Summary:**

The paper has good originality and high significance in the topic of retrieval-augmented generation. The paper lacks some analyses, causing a possibility that the paper's claim is not verified well, lowering the quality. The clarity is fine. See "Reasons To Accept" and "Reasons To Reject" below for details.

**Questions To Authors:**

- Section 2.1: I assume Y_new can significantly (semantically) differ from Y_origin, based on Section 4.4.  If so, how does one know when to stop the iterative query refinement annotation process and make Y_new?  This point would be better clarified if the paper gave an example and presented details of the answer template shown in Figure 2.
- Figure 3: Section 4.3 says the confidence-based sampling strategy generally excels for single-hop QA tasks, but actually Figure 3 shows that the ensemble-based strategy performs best across single-hop and multi-hop tasks, except for OBQA. What is the reason why this anomaly (i.e., the confidence-based strategy outperforming the ensemble-based one) happens with OBQA?
- Appendix A.1: More detailed dataset statistics should be provided.  It would be interesting to see the number of occurrences of each query refinement and the average number of refined queries in each QA pair.
- Section 2.3: However, using different query to -> However, using different *queries* to
- Section 2.3: ... which leads to different final answer. -> ... which leads to different final *answers*.
- Section 2.3: We ensemble the final results by select the final results ... -> We ensemble the final results by selecting the final results ...

**Reasons To Accept:**

1) The empirical observations support the superiority of the proposed method well. They show that the proposed method outperforms state-of-the-art models in single-hop and multi-hop QA tasks (Self-RAG-7B and GPT-3.5-turbo, respectively).
2) The paper conducts extensive comparison experiments using four datasets for each of single-hop QA and multi-hop QA.
3) Although I see a few typos, the paper is well organized, making it easy to follow the content.

**Reasons To Reject:**

1) The contribution by each of learned capabilities (e.g., query decomposition) is not adequately analyzed. In particular, I suspect the contribution by query disambiguation might be minimal because ambiguous QA data (ASQA) occupies only 10% of the constructed dataset, according to Figure 5. The contribution could be more convincingly justified if it showed statistics of how many occurrences of query refinement types exist in the dataset and how many times each of those refinement actions was chosen by LLM during inference. See also the question below.
2) The clarification process by the proposed query disambiguation could be inefficient because a large number of disambiguated queries is potentially possible, especially if a disambiguated aspect is at the instance level.  For example, if a question is "What is the length of a U.S. president's term?", the answer depends on a particular U.S. president.  Then, the proposed method could end up generating numerous queries "What is the length of the term for the U.S. president Joe Biden?", "What is the length of the term for the U.S. president Donald Trump?", etc.  If numerous disambiguated queries are generated, the proposed method will lack runtime efficiency, and LLM will be likely to generate erroneous final answers.
3) No error analysis is provided. The paper would be more insightful for future work if it showed an analysis of what errors are dominantly produced from each of the query refinement actions.

---

> ### Author Rebuttal · Authors · 2024-05-31
>
> Thanks for your insightful comments!
>
> **1. More Details on the Effect of Different Actions**
>
> Regarding the concern about the effect of different actions, we have complemented the new experiment as follows, and will provide case study to support the error analysis in our next version. Note that the newly provided results are based on our newly trained llama3 8b model.
>
> - **The ratio of different actions under different sampling strategies of 3 tasks.**
>
>
> |       | Decompose | Disambiguate | Rewrite | Direct |
> |----------------|-----------|--------------|---------|--------|
> | **Confidence** | 0.46      | 0.35         | 0.19    | 0.03   |
> | **Ensemble**   | 0.34      | 0.32         | 0.34    | 0.01   |
> | **PPL**        | 0.53      | 0.37         | 0.10    | 0.00   |
>
> - **F1 score  for deliberately using the specific action for all instances of 3 tasks.**
>
> |    | HotpotQA | 2Wiki | Musique | Avg     |
> |--------------------------|----------|-------|---------|---------|
> | **Rewrite**              | 74.8 | 59.2| 46.2 | 60.1 |
> | **Decompose**            | 73.2 | 58.7 | 48.4 | 60.1 |
> | **Disambiguate**         | 72.6| 60.3 | 45.0 | 59.3|
> | **With Sampling Strategies** | 74.2     | 59.3  | 48.1    | 60.5    |
>
> The findings are as follows:
>
> - Although the disambiguation process accounts for only 10% of our training data, the proportion of different action selections is almost balanced.
>
> - Simply choosing the same action might not yield the best performance compared to the score with sampling strategies.
>
> **2. Inefficiency of the Disambiguation Process**
>
> The inefficiency arises from generating multiple queries and retrieving each piece of evidence, which increases inference latency. This is not a drawback of our method alone; multiple strong baselines, such as Self-Ask and ReAct, also generate follow-up questions multiple times to retrieve the necessary information. Regarding the numerous disambiguated queries generated, our trained model is designed to answer all of them, covering all generated queries.
>
>  **3. Clarification of Iterative Query Refinement Annotation Process**
>
> We use the generated queries to retrieve documents from an external database, then concatenate them to regenerate Y_new, which happens only once. The intuition behind this process is that Y_new should be conditioned on all generated queries to cover all the queries. During inference, the model can decide to generate a refined query or choose to terminate and generate the final answer.

---

> > ### Comment · Reviewer_PYYL · 2024-06-06
> > **Re: Rebuttal by Authors**
> >
> > Thank you for the response. It's good to see the action distribution is balanced. I'll keep my original evaluation.

---

### Decision · Program_Chairs · 2024-07-10

**Decision:**

Accept

**Comment:**

This paper proposes RQ-RAG to train (a relatively small) LM to better solve QA tasks, including explicitly rewriting the query, decomposing the task, and disambiguating the queries. The authors showed significant performance boosts across both single-hop and multi-hop QA datasets over the LLama2 7B model.

Pros:
- All reviewers agree that the improvements are quite significant and the experiments are fairly comprehensive.
- The method is well motivated and the framework is carefully designed.

Cons:
- Lack of baselines on multi-hop datasets: some reviewers point out the lack of baselines for multi-hop QA, e.g., many existing prompting methods aim at decomposing the multi-hop questions and answer them better, least-to-most, self-ask, etc. The authors address this to some extent in their rebuttal, but those baselines should be added to the main paper to be comprehensive.
- The authors claim existing decomposition methods like self-ask don't work well with small models. However, the proposed RQ-RAG used ChatGPT for data collection, so it is not a very fair comparison for self-ask on LLama2-7B only vs RQ-RAG.
- Contribution of each component is not super clear, better ablation experiments should be added.

Overall I'm leaning accept given the strong empirical gains but hope the authors can add more comprehensive baselines and ablations in the revision.